# Structural Assignment of the Product Ion Generated from a Natural Ciguatoxin-3C Congener, 51-Hydroxyciguatoxin-3C, and Discovery of Distinguishable Signals in Congeners Bearing the 51-Hydroxy Group

**DOI:** 10.3390/toxins16020089

**Published:** 2024-02-06

**Authors:** Ryogo Ukai, Hideaki Uchida, Kouichi Sugaya, Jun-ichi Onose, Naomasa Oshiro, Takeshi Yasumoto, Naoki Abe

**Affiliations:** 1Department of Nutritional Science and Food Safety, Graduate School of Applied Bioscience, Tokyo University of Agriculture, 1-1-1 Sakuragaoka, Setagaya, Tokyo 156-8502, Japan; 10821002@nodai.ac.jp (R.U.); k3sugaya@nodai.ac.jp (K.S.); j1onose@nodai.ac.jp (J.-i.O.); 2Japan Customer Service Organization, Agilent Technologies Japan, Ltd., 9-1 Takakura-machi, Hachioji, Tokyo 192-8510, Japan; hideaki_uchida@agilent.com; 3National Institute of Health Sciences, 3-25-26 Tonomachi, Kawasaki 210-9501, Japan; n-oshiro@nihs.go.jp; 4Tama Laboratory, Japan Food Research Laboratories, 6-11-10 Nagayama, Tama, Tokyo 206-0025, Japan; yasumotot@gmail.com

**Keywords:** ciguatoxins (CTXs), 51-hydroxyciguatoxin-3C (51-hydroxyCTX3C), LC-APCI-QTOFMS, charge-remote fragmentation (CRF), polycyclic ethers

## Abstract

Ciguatoxins (CTXs) stand as the primary toxins causing ciguatera fish poisoning (CFP) and are essential compounds distinguished by their characteristic polycyclic ether structure. In a previous report, we identified the structures of product ions generated via homolytic fragmentation by assuming three charge sites in the mass spectrometry (MS)/MS spectrum of ciguatoxin-3C (CTX3C) using LC-MS. This study aims to elucidate the homolytic fragmentation of a ciguatoxin-3C congener. We assigned detailed structures of the product ions in the MS/MS spectrum of a naturally occurring ciguatoxin-3C congener, 51-hydroxyciguatoxin-3C (51-hydoxyCTX3C), employing liquid chromatography/quadrupole time-of-flight mass spectrometry with an atmospheric pressure chemical ionization (APCI) source. The introduction of a hydroxy substituent on C51 induced different fragmentation pathways, including a novel cleavage mechanism of the M ring involving the elimination of 51-OH and the formation of enol ether. Consequently, new cleavage patterns generated product ions at *m*/*z* 979 (C_55_H_79_O_15_), 439 (C_24_H_39_O_7_), 149 (C_10_H_13_O), 135 (C_9_H_11_O), and 115 (C_6_H_11_O_2_). Additionally, characteristic product ions were observed at *m*/*z* 509 (C_28_H_45_O_8_), 491 (C_28_H_43_O_7_), 481 (C_26_H_41_O_8_), 463 (C_26_H_39_O_7_), 439 (C_24_H_39_O_7_), 421 (C_24_H_37_O_6_), 171 (C_9_H_15_O_3_), 153 (C_9_H_13_O_2_), 141 (C_8_H_13_O_2_), and 123 (C_8_H_11_O).

## 1. Introduction

Ciguatera fish poisoning (CFP) affects over 20,000 individuals annually, predominantly in tropical and subtropical regions [1]. Ciguatoxins (CTXs), primarily produced by the epiphytic dinoflagellate *Gambierdiscus* spp., stand as the main agents causing CFP [1,2]. These toxins accumulate in fish through the food chain and undergo oxidation via fish metabolism [1,3]. CTXs exhibit ladder-like polycyclic ether structures, with variations in their structural skeletons based on their origin. While the detailed chemical structure of the Indian Ocean family remains unclear, the Pacific family is categorized into CTX3C and CTX1B types, with over 20 congeners’ structures determined so far (Figure 1) [1,4]. The specific structural differences are that CTX3C types have no substituted ring A or eight-membered ring E, while CTX1B types possess a C4 substituent on ring A and a seven-membered ring E. They are crucial compounds in terms of food hygiene. However, studies aiming to advance toxicity assessments based on their chemical structures are lacking because obtaining sufficient amounts for NMR analysis is quite difficult [1]. Therefore, isolated new CTX analogs have had their structures elucidated through fast atom bombardment (FAB) mass spectrometric analysis [1,5,6].

The ionic charges situated near the end of the molecule induce charge-remote fragmentation (CRF), enabling predictable spectra and direct interpretation. Despite CTXs lacking ionic charges, the presence of two nearby oxygen atoms, such as the ether oxygen of ring A and 7-OH, and the spiroketal oxygen atoms in CTX3C, interact to set Na^+^ at their sites, resulting in the generation of product ions akin to those in CRF. The structure of most homologues could be identified by employing the fragmentation pattern of known structures as a template [1].

Recently, liquid chromatography-tandem mass spectrometry (LC/MS/MS) with electrospray ionization (ESI) as the ion source has been extensively used to detect and quantify CTXs [7,8,9]. Despite their extremely low levels in ciguatera fishes, LC/MS/MS’s stability of the precursor [M + Na]^+^ ion facilitates detection and quantification, with a sensitivity at 0.01 μg/kg for CTX1B equivalent in FDA guidance levels [3,9]. However, this method is not suitable for assigning product ion structures, necessitating structural analysis to obtain detailed fragment ions as MS/MS analysis selects precursor ions such as [M + H]^+^ or [M + NH_4_]^+^ [10,11].

Despite the continuous dehydration of CTXs in MS/MS spectra, no studies have reported on the dehydration mechanism or structures of product ions [4,5,6,10,11,12]. In our previous study, we identified the chemical structures of product ions generated from three types of [M + H]^+^, with respective charge sites, of synthesized CTX3C (commercially available) using liquid chromatography/quadrupole time-of-flight mass spectrometry with atmospheric pressure chemical ionization (APCI). This method did not reveal an intensive [M + Na]^+^ ion source (LC-APCI-QTOFMS). Particularly, we elucidated the mechanism of generating characteristic sequential dehydration from various ladder-shaped polycyclic ethers [13,14]. This understanding arose from the ether oxygen removing two hydrogens from the carbons on either side, resulting in double bond conjugation stabilization with a double bond similarly produced in a neighboring ring. These findings marked the initial report unveiling the structure of product ions generated from CTX3C in the LC-MS/MS spectrum [15].

In this study, we assigned the structures of the product ions generated from the [M + H]^+^ of naturally occurring 51-hydroxyCTX3C [1] based on CRF [16,17]. Additionally, distinct fragmentation pathways were observed in CTX3C congeners bearing a 51-hydroxy group, aligning with CRF and revealing a novel cleavage mechanism of the M ring involving the elimination of 51-OH and the formation of enol ether. We anticipate that comparing these results with those obtained from CTX3C will offer valuable insights for the structural analysis of unidentified analogues.

## 2. Results and Discussion

Figure 2 exhibits the LC/MS/MS spectrum displaying product ions from [M + H]^+^ of 51-hydroxyCTX3C, acquired using optimized MS conditions derived from previous CTX3C measurements [15]. A modified collision energy (CE = 30 eV) was employed. Product ion assignment from [M + H]^+^ at *m*/*z* 1039.5647 (C_57_H_83_O_17_, calculated 1039.5630) was established by comparing it with those generated from CTX3C (Appendix A). Unlike CTX3C, 51-hydroxyCTX3C, while neutral like CTX3C, contains multiple oxygen atoms with lone pairs of electrons. Two closely located oxygen atoms chelate protons, acting as charge sites, led us to postulate three similar charge sites within the molecule: A between the ether oxygen in ring A and the 7-OH in ring B, B between spiroketal oxygen atoms of rings L and M, and C between the 7-OH of ring B and the ether oxygen of ring C.

Product ions from 51-hydroxyCTX3C generated based on charge site A exhibited CRF-induced cleavage similar to CTX3C (Figure 3a). The product ion at *m*/*z* 907.4851 (C_51_H_71_O_14_, calculated 907.4844, A1) underwent three dehydration steps, yielding *m/z* 889.4670 (C_51_H_69_O_13_, calculated 889.4738, A2), *m*/*z* 871.4534 (C_51_H_67_O_12_, calculated 871.4633, A3), and *m*/*z* 853.4475 (C_51_H_65_O_11_, calculated 853.4527, A4) in Appendix A. Moreover, ions resulting from sequential dehydration from rings F and E produced *m*/*z* 523.2692 (C_31_H_39_O_7_, calculated 523.2696, A6) and *m*/*z* 505.2612 (C_31_H_37_O_6_, calculated 505.2590, A7) from the product ion at *m*/*z* 541.2794 (C_31_H_41_O_8_, calculated 541.2801) attributed to structure A5 in Appendix A. Additionally, the ion at *m*/*z* 109.0652 (C_7_H_9_O, calculated 109.0653, A8) was detected (Figure 3a and Appendix A).

Furthermore, a new fragmentation pathway leading to an ion at *m*/*z* 979.5385 (C_55_H_79_O_15_, calculated 979.5419, A9), derived from CRF at charge site A, emerged via fission in ring M (Figure 3a,b and Appendix A). The resulting structure A9, with a double bond C49=C50, resulted from the cleavage between C50 and oxidized C51 conjugated with the α-cleavage of the ether bond C49-O. The characteristic sequential dehydration of the polycyclic ethers in rings K through J produced ions at *m*/*z* 961.5296 (C_55_H_77_O_14_, calculated 961.5313, A10), 943.5180 (C_55_H_75_O_13_, calculated 943.5208, A11), and 925.5189 (C_55_H_73_O_12_, calculated 925.5102, A12), halted by the presence of a methyl group at C30 where a lack of hydrogen withdrawal occurred, as shown in Figure 3b. Considering the existence of this new fragment pathway, it is indicated that 51-OH might facilitate the occurrence of the characteristic conversion or cleavage of ring M (Figure 3b).

The preference of 51-OH for selective dehydration was confirmed by comparable fragmentation patterns to those of CTX3C derived from charge site B (Figure 4 and Appendix A). Bond cleavage in ring G led to ions at *m*/*z* 509.3077 (C_28_H_45_O8, calculated 509.3114, B1), 491.3017 (C_28_H_43_O_7_, calculated 491.3009, B2), 473.2848 (C_28_H_41_O_6_, calculated 473.2903, B3), 455.2749 (C_28_H_39_O_5_, calculated 455.2797, B4), and 437.2653 (C_28_H_37_O_4_, calculated 437.2692, B5) (Figure 4a, Appendix A). In contrast, bond cleavages in rings G and H produced product ions at *m*/*z* 481.2808 (C_26_H_41_O_8_, calculated 481.2801, B6), 463.2709 (C_26_H_39_O_7_, calculated 463.2696, B7), 445.2574 (C_26_H_37_O_6_, calculated 445.2590, B8), 427.2462 (C_26_H_35_O_5_, calculated 427.2484, B9), 409.2342 (C_26_H_33_O_4_, calculated 409.2379, B10), and 391.2247 (C_26_H_31_O_3_, calculated 391.2273, B11) (Figure 4a, Appendix A).

Sequential dehydration, commonly observed in cyclic polyether structures like CTXs, was evident [4,5,6,10,11,12,13]. While CTX3C exhibited seven steps of sequential dehydration stopping at the methyl group on C30 of ring G, 51-hydroxyCTX3C demonstrated signals for eight steps of dehydration (Appendix A). This difference might be attributed to the removal of 51-OH along with 50-H, favoring the formation of the double bond C50=C51, followed by the sequential formation of polycyclic ether dehydration. Consequently, eight product ions at *m*/*z* 1021.5508 (C_57_H_81_O_16_, calculated 1021.5525, B16-2), 1003.5413 (C_57_H_79_O_15_, calculated 1003.5419, B17-2), 985.5294 (C_57_H_77_O_14_, calculated 985.5313, B18-2), 967.5161 (C_57_H_75_O_13_, calculated 967.5208, B19-2), 949.5080 (C_57_H_73_O_12_, calculated 949.5102, B20-2), 931.4953 (C_57_H_71_O_11_, calculated 931.4996, B21-2), 913.4797 (C_57_H_69_O_10_, calculated 913.4891, B22-1), and 895.4706 (C_57_H_67_O_9_, calculated 895.4785, B23) were detected as product ions, displaying the characteristic continuous dehydration state from the A ring observed in CTXs (Figure 5 and Appendix A). On the other hand, it was expected that there were also product ions induced by CRF through a pathway similar to that of CTX3C. The product ions generated by the pathway are also shown in Figure 5 (from B16-1 to B22-1). In this case, if dehydration of 51-OH occurs according to CRF, dehydration may occur at B-22-1 to form B23-2. All fragmentations, except *m*/*z* 239, were confirmed in 51-hydroxyCTX3C, indicating a high probability of 51-OH dehydration (Figure 4, Figure 5 and Appendix A).

The second novel fragmentation pathway, B, generated from charge site B (at ring L-M), yielded ion B24 at *m*/*z* 439.2684 (assigned as C_24_H_39_O_7_, calculated 439.2696), featuring a double bond C33=C34 resulting from the cleavage between C32 and C33 conjugated with the α-cleavage of C34-O based on CRF (Figure 6 and Appendix A). Although successive dehydration steps were observed at *m*/*z* 421.2626 (C_24_H_37_O_6_, calculated 421.2590, B25-2, including B25-1) and 403.2456 (C_24_H_35_O_5_, calculated 403.2484, B26-2, including B26-1) (Appendix A), the signal strength of *B25* was higher than that of B24, suggesting that the removal of 51-OH along with 50-H to form a double bond might be more predominant than the sequential dehydration of the polycyclic ethers (Figure 7b and Appendix A). In the sequential dehydration of all these product ions originating from charge site B, the ions that were expected to undergo dehydration with a 51-hydroxy substituent were observed to exhibit stronger intensities than those possessing an intact M ring with 51-OH.

The MS/MS spectrum of CTX3C displays two simple product ions at *m*/*z* 125 and 155 in the low MS range (Figure 7a), characteristic signals that correspond to two representative Pacific CTX types (CTX4A and CTX3C in Figure 1), both possessing a spiroketal structure between rings L and M. Conversely, the spectrum of 51-hydroxyCTX3C revealed more complex signals in the low MS range (Figure 7a). The ions at *m*/*z* 171.1038 (C_9_H_15_O_3_, calculated 171.1021, B12) and *m*/*z* 141.0904 (C_8_H_13_O_2_, calculated 141.0916, B14) corresponded to product ions associated with the 51-hydroxy group at *m*/*z* 155 and 125, respectively, as assigned from CTX3C (Appendix A). Additionally, at ring M with 51-OH, the fragmentation was expected to remove 51-OH along with 50-H, forming a stable double bond C50=C51 (Figure 7b), leading to the ions of *m*/*z* 153.0920 (C_9_H_13_O_2_, calculated 153.0916, B13) and 123.0800 (C_8_H_11_O, calculated 123.0810, *B15*). To assign the signal at *m*/*z* 135.0794 (C_9_H_11_O, calculated 135.0810, B27), alternative paths from B12 via the ion productions of B13-2 rather than B-13 were assumed (Figure 8a). When 51-OH is eliminated from the M ring, a hydrogen atom is extracted from C50 to form a C50=C51 double bond, and another hydrogen atom is removed from the C52 position to produce a C51=C52 double bond. While C50=C51 forms an energetically stable structure, C51=C52 becomes an energetically unstable enol ether structure; therefore, if the reaction progresses further, the M ring will cleave. Subsequently, a triple bond is introduced between C51 and C52 through a dehydration reaction, generating the desired product ion (B27) (Figure 8a). On the other hand, the product ion at *m*/*z* 149.0961 (C_10_H_13_O, calculated 149.0966), structurally assigned to B28, originated from the cleavage of the C44-C45 bond accompanied by α-cleavage of C46-(O). It was predicted that the introduction of C51≡C52 occurred through a dehydration reaction via an enol ether, similar to B27 (Figure 8b). Ions predicted to be generated in Figure 8a,b but not detected in the spectrum are energetically unstable; therefore, they might quickly transform into stable ions. The *m*/*z* 149 ion is a crucial product ion utilized for quantification in MS/MS, common to P-CTXs that feature a hydroxyl group on the same carbon position in the M ring, such as CTX1B and 51-hydroxyCTX3C. This result not only clarifies its structure for the first time but also provides a novel cleavage mechanism of the M ring, involving the elimination of 51-OH and the formation of an enol ether [7]. Although the formation and structure of B27 and B28 are thought to be rational, the reaction mechanism from, for example, B13-2 to B13-3, is not yet clear.

The ion at *m*/*z* 115.0756 (C_10_H_13_O, calculated 115.0759, B29) resulted from simultaneous cleavages in rings L and M. The cleavage of bonds at C46-C47 and C45-O in ring L formed an oxetane ring, while the fission of bonds at C50-C51 and C52-O in ring M generated an oxirane ring, leading to the formation of ion (B29) (Figure 8c). This fragmentation, involving the M ring and the 51-hydroxy substituent, differed from the 51-OH dehydration pattern.

The novel ions assigned from charge site B in 51-hydroxyCTX3C suggest that the presence of the 51-hydroxyl substituent triggers various fragmentation patterns, including the opening of the M ring, resulting in complex signals in the low MS range.In this study, the fragmentation observed from charge site C for the [M + H]+ of CTX3C was not detected in 51-hydroxyCTX3C [13]. At charge site C, the proton might have remained uncharged, or fragmentation might have been inhibited due to the hydroxy substituent on C51. The product ions identified in this study offer crucial information for structural analysis, particularly in identifying the presence of a hydroxyl group at C51 of the M ring in an unknown CTX congener. Furthermore, the analysis of signals in the low MS region, which could not be attributed this time, is expected to yield further significant findings.

## 3. Conclusions

Based on the analysis of mechanisms generating product ions from the [M + H]^+^ of 51-hydroxyCTX3C using LC-APCI-QTOFMS, this study reveals that the hydroxyl substituent on C51 induces the creation of specific product ions, such as *m*/*z* 979.5385 (C_55_H_79_O_15_, A9) and 439.2684 (C_24_H_39_O_7_, B26), utilizing new cleavage patterns. These result in characteristic product ions, notably at *m*/*z* 509.3077 (C_28_H_45_O_8_, B1), 491.3017 (C_28_H_43_O_7_, B2), 481.2808 (C_26_H_41_O_8_, B6), 463.2709 (C_26_H_39_O_7_, B7), and others, formed by dehydrolyzing 51-OH and 50-H to create a double bond C50=C51. Furthermore, novel ions such as *B27* at *m*/*z* 135.0794 (C_9_H_11_O), B28 at *m*/*z* 149.0961 (C_10_H_13_O), and B29 at *m*/*z* 115.0756 (C_6_H_11_O_2_) from charge site B in 51-hydroxyCTX3C suggest varied fragmentation patterns due to the presence of the 51-hydroxyl substituent, including the opening of the M ring, leading to complex signals in the low MS range.

These findings enable the elucidation of the chemical structure of 51-hydroxyCTX3C derivatives present in extremely small natural amounts without purification, using only crude samples. Similar to CTX3C, it was assumed that 51-hydroxyCTX3C has three charge sites. While confirmations of fragmentation from charge sites A and B are evident in 51-hydroxyCTX3C, no observation of fragmentation from charge site C occurs in CTX3C. This outcome suggests that substituted hydroxyl groups near CTX molecule termini impact the formation of charge sites that fix ions between adjacent oxygen atoms.

Analyzing pathway analyses of product ions using LC/APCI-QTOFMS spectra for known homologues with different hydroxyl groups near molecular terminals, such as CTX1B, 2,3-dihydroxyCTX3C, and C-CTX-1, may offer insights into structures of minor CTX congeners in crude samples. By further exploring MS/MS spectra of 2,3-dihydroxyCTX3C and 2,3,51-trihydroxyCTX3C, significant data associations can be drawn, reflecting the characteristics of both CTX3C and 51-hydroxyCTX3C. This research advancement paves the way for identifying unknown CTX3C derivatives in nature.

## 4. Materials and Methods

Acetonitrile, methanol, and distilled water (LC-MS or analytical-grade chemicals) for LC-MS, along with CTX3C, were acquired from FUJIFILM Wako Pure Chemicals Corporation (Osaka, Japan). Formic acid and ammonium formate (LC-MS grade) were sourced from Sigma–Aldrich (St. Louis, MO, USA).

The experiment utilized liquid chromatography-QTOF mass spectrometry equipped with an APCI source (LC-APCI-QTOFMS).

All procedures were conducted using an Agilent 1260 LC binary system (Waldbronn, Germany), comprising a binary pump integrated with a vacuum degasser, an autosampler, and a thermal compartment. This system was coupled to an Agilent 6530 QTOF LC/MS system (Santa Clara, CA, USA) equipped with an APCI source. A reversed-phase C18 analytical column (Poroshell 120 EC-C18; 4.6 mm i.d. × 100 mm, 2.7 μm particle size; Agilent Technologies) was employed, eluting at 1.0 mL/min with solvents A: water containing 0.1% formic acid and 5 mM ammonium formate and B: methanol. Isocratic conditions at 10% solvents A and B were maintained for 10 min. The column temperature was set at 40 °C, with an injection volume of 50 μL (2 ng/mL).

For liquid chromatography high-resolution mass spectrometry (LC/HRMS) and LC/MS/MS experiments, positive ion mode was employed within the mass range *m*/*z* 200–1300 in MS mode and *m*/*z* 100–1300 in MS/MS mode. Operational parameters included a capillary voltage of 5000 V, nebulizer pressure of 40 psi, N_2_ gas flow of 10 L/min, drying gas temperature of 300 °C, vaporizer temperature of 250 °C, skimmer voltage of 65 V, Corona current of 40 mA, and fragmentor voltage of 250 V. Continuous automated internal mass calibration and real-time automated mass correction for all spectra were performed using known internal reference masses (*m*/*z* 121.0508, 1221.9810) at a flow/isocratic pump in MS modes. Additionally, nebulizers introducing a calibrating solution at low flow containing internal reference mass compounds (*m*/*z* 322.0481 and 1221.9810) were utilized for constant automated internal mass calibration.

Furthermore, MS/MS experiments (precursor ion; [M + H]^+^ *m*/*z* 1039.5600) were conducted in positive ion mode within the mass range *m*/*z* 100–1300.The parameters included a collision energy (CE) of 30 eV, an MS acquisition rate of 2 spectra/s, an MS/MS acquisition rate of 1 spectrum/s, and a target isolation width of 4 *m*/*z* units. Data analysis was performed using the Agilent MassHunter software (version B.6.00). The QTOFMS data used for assignments had a maximum tolerance of 10 ppm, unless otherwise specified.

## Figures and Tables

**Figure 1 toxins-16-00089-f001:**
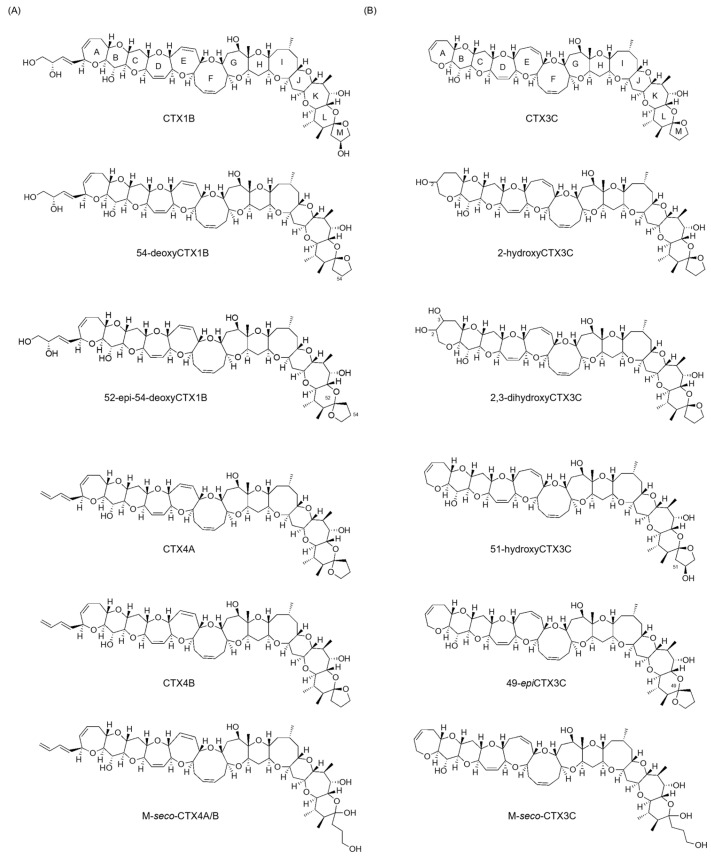
Structures of (**A**) CTX1B and (**B**) CTX3C congeners.

**Figure 2 toxins-16-00089-f002:**
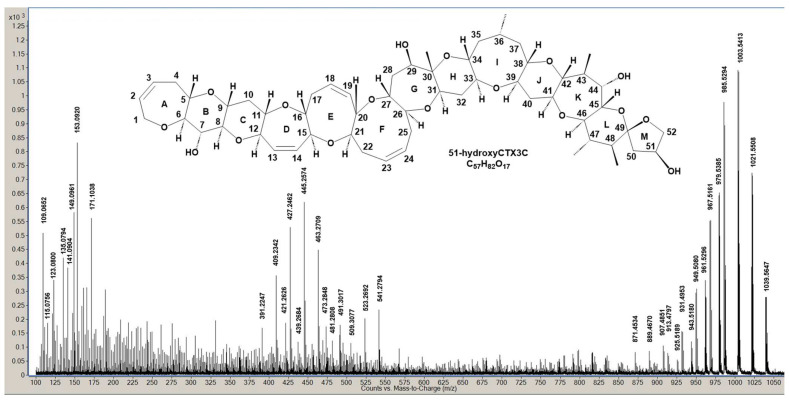
APCI-MS/MS product ion spectrum of 51-hydoxyCTX3C [M + H]^+^ as a precursor ion.

**Figure 3 toxins-16-00089-f003:**
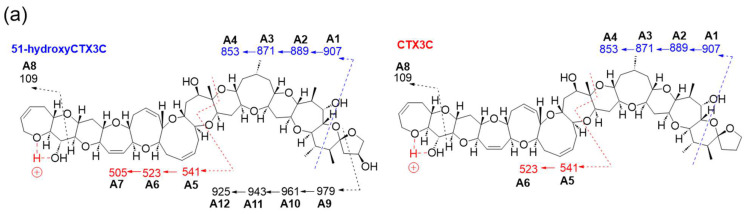
(**a**) Comparison of fragmentation patterns induced in charge site A of 51-hydroxyCTX3C and CTX3C. (**b**) Structures of new assigned product ions (*m*/*z* 979→) from 51-hydroxyCTX3C induced by charge site A. [The figure in the dashed frame shows detailed cleavage mechanism. The parts to emphasize are shown in red].

**Figure 4 toxins-16-00089-f004:**
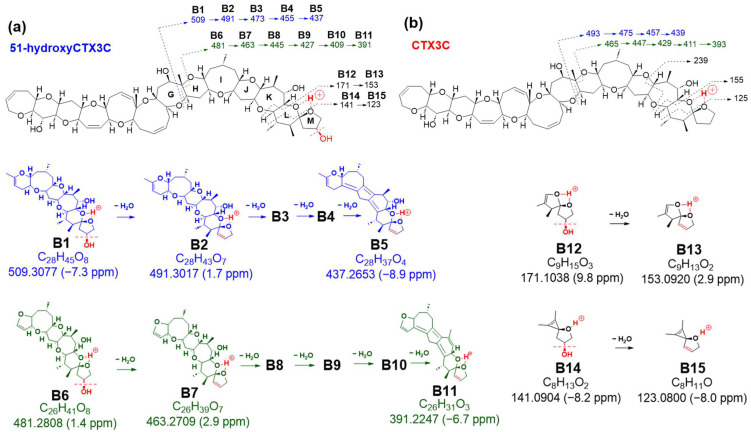
Comparison of fragmentation patterns induced in charge site A of 51-hydroxyCTX3C and CTX3C. [The parts to emphasize are shown in red. The blue and green colors represent each fragment series].

**Figure 5 toxins-16-00089-f005:**
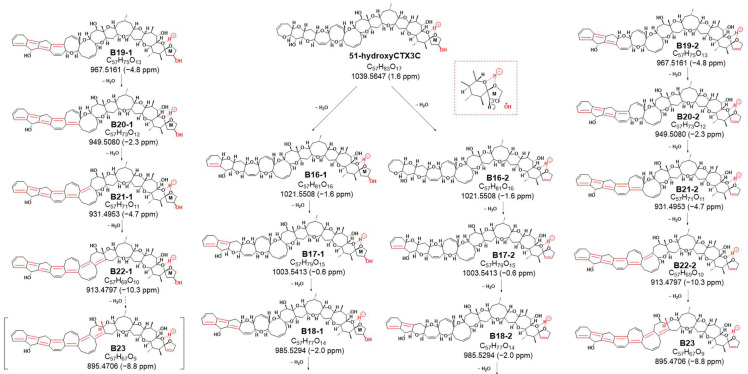
Product ions of 8 sequential dehydrations in the MS/MS spectrum of 51-hydroxyCTX3C. [The parts to emphasize are shown in red].

**Figure 6 toxins-16-00089-f006:**
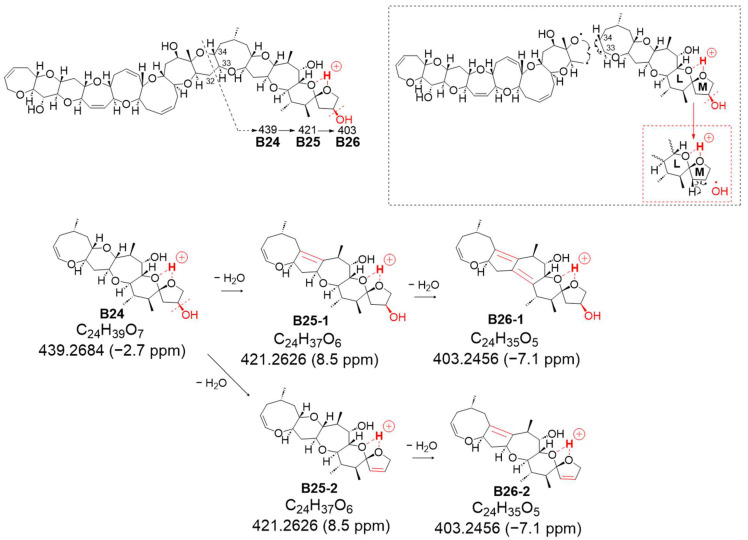
Structures of new assigned product ions (*m*/*z* 439→) from 51-hydroxyCTX3C induced by charge site B. [The figure in the dashed frame shows detailed cleavage mechanism. The parts to emphasize are shown in red].

**Figure 7 toxins-16-00089-f007:**
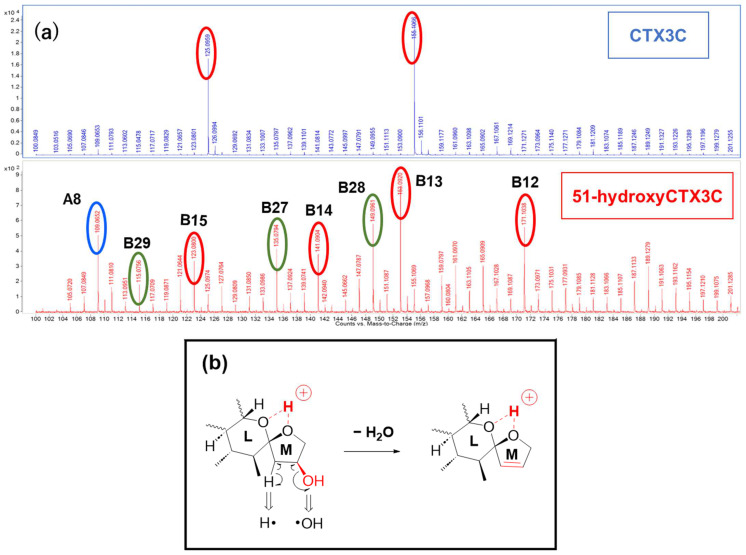
(**a**) Comparison signals of 51-hydroxyCTX3C and CTX3C in the low MS range (*m*/*z* 100–200) of the MS/MS spectrum. [The signals circled in red are the ions generated by the same fragmentation pattern as CTX3C. The blue shows the same product ion from CTX3C (see Figure 3a). The green are the newly observed product ions of 51-hydroxyCTX3C.] (**b**) Fragmentation pattern of ring M induced in charge site B of 51-hydroxyCTX3C. [The parts to emphasize are shown in red].

**Figure 8 toxins-16-00089-f008:**
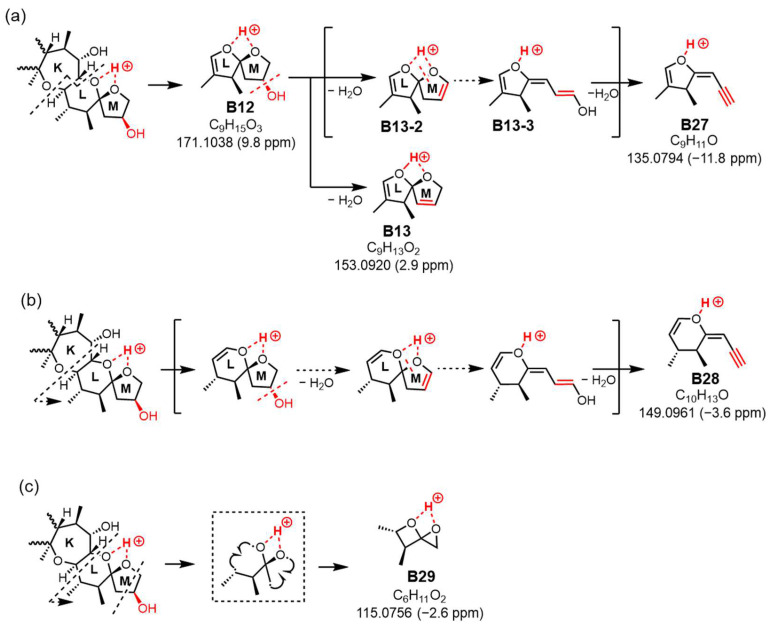
Generate mechanism of product ion at *m*/*z* 135 (**a**), *m*/*z* 149 (**b**), and *m*/*z* 115 (**c**). The parts to emphasize are shown in red.

## Data Availability

The data presented in this study are available on request from the corresponding authors.

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
