# Peer review of "Structural Assignment of the Product Ion Generated from a Natural Ciguatoxin-3C Congener, 51-Hydroxyciguatoxin-3C, and Discovery of Distinguishable Signals in Congeners Bearing the 51-Hydroxy Group"

_toxins, 2024, doi:10.3390/toxins16020089_

Round 1

Reviewer 1 Report

Comments and Suggestions for Authors

Ciguatoxin and its congeners are very important marine toxins. Authors identified the product ions of 51-hydroxyCTX3C by APCI-MS/MS. Those results will lead to structural elucidation of ciguatoxin analogs and help ciguatera researchers. Therefore, I recommend this manuscript is to be published in Toxins after minor revision.

 1. First of all, 51-hydroxyciguatoxin in the title is spelled incorrectly.

2. The resolution of MS spectra is low. It is difficult to read the numbers.

3. In Figure 5, Dehydration of 51-OH occurs at the early stage. On the other hand in Figure 6, dehydration of 51-OH occurs after cleavage of C32-C33 and O-C34. Why do they occur by different mechanism?

4. Figure 7 (b).  The caption is wrong. CTX3C does not have 51-OH. So dehydration does not occur in CTX3C.   

Author Response

Thank you for your peer review to our manuscript.

We have revised the manuscript ID: toxins-2813151 on the basis of the Referee’s comments.

We look forward to a publication of our manuscript in the toxins.

Reviewer 1

General Points.

Comment 1.

 First of all, 51-hydroxyciguatoxin in the title is spelled incorrectly.

Response 1.

(Title) “51-Hydroxycigatoxin-3C” => “51-Hydroxyciguatoxin-3C”

Comment 2.

The resolution of MS spectra is low. It is difficult to read the numbers.

Response 2.

We apologize for the inconvenience. However, according to the specifications of the manuscript, the image size and resolution will be maximum in this manuscript.

Therefore, we have attempted to provide a maximally expanded spectrum of the entire spectrum in supplementary materials (Figure S11).

Comment 3.

In Figure 5, Dehydration of 51-OH occurs at the early stage. On the other hand in Figure 6, dehydration of 51-OH occurs after cleavage of C32-C33 and O-C34. Why do they occur by different mechanism?

Response 3.

In this paper, Figures 5 and 6 show the induction of product ions that are expected to be newly generated due to the introduction of an OH group at C-51, but this does not mean that 51-OH is preferentially dehydrated in Figure 5. It is expected that there will also be product ions induced by CRF in the same way as CTX3C. To avoid misunderstandings, we have changed Figures 5 and 6 as follows.

Figure 5 shows two fragmentation pathways: one in which dehydration of 51-OH occurs in the first step (B16-2 to B23-2), and one in which continuous dehydration progresses while 51-OH continues to exist (B16-1 to B22-1). Furthermore, in the latter case, if dehydration of 51-OH occurs according to CRF, dehydration may occur at B-22-1 to form B23. Figure 6 also shows two fragmentation pathways (B24 to B26-1 or B26-2).

Additionally, we added the following sentences.

(lines 148 to 152)

“On the other hand, it was expected that there are also product ions induced by CRF through a pathway similar to that of CTX3C. The product ions generated by the pathway were also shown in Figure 5 (B16-1 to B22-1). In this case, if dehydration of 51-OH occurs according to CRF, dehydration may occur at B-22-1 to form B23.”

Comment 4.

Figure 7 (b). The caption is wrong. CTX3C does not have 51-OH. So dehydration does not occur in CTX3C.

Response 4.

[Figure 7 (b) title] “51-hydroxyCTX3C and CTX3C” => “51-hydroxyCTX3C”

Reviewer 2 Report

Comments and Suggestions for Authors

Dear Editor, Greetings.

I report that I have read in detail the article entitled "Structural Assignment of the Product Ion Generated from a Natural Ciguatoxin-3C Congener, 51-Hydroxycigatoxin-3C, and Discovery of Distinguishable Signals in Congeners Bearing the 51-Hydroxy Group”.

Very well presented article and detail regarding the results from Natural Ciguatoxin-3C Congener.

I have only minor comments which I detail below:

Fig 1 & 8: Please be more precise in the description of the figures.

Page 3 line 71-77: Please reword this paragraph and move the comments linked to figure 2 to section 2 (Results & Discussion). Additionally, move the figure to a more appropriate section of the text and include with more precision and detail the objective of the article.

Comments on the Quality of English Language

Moderate editing of English language required.

Author Response

Thank you for your peer review to our manuscript.

We have revised the manuscript ID: toxins-2813151 on the basis of the Referee’s comments.

We look forward to a publication of our manuscript in the toxins.

Reviewer 2

Comment 1.

Fig 1 & 8: Please be more precise in the description of the figures.

Response 1.

Regarding Figure 1,

We added the following sentence.

(lines 36 to 39)

“The specific structural differences are that CTX3C types have no substituted ring A, and eight-membered ring E while CTX1B types possess a C4 substituent on ring A, and seven-membered ring E.”

Regarding Figure 8,

Although the generation and structure of the ions (B27 and B28) are considered reasonable, the reaction mechanism from B13-2 to B13-3 is not clear, so the corresponding parts of Figures 8(a) and (b) are Changed the representation from a solid arrow to a dashed line.

We added the following sentence.

(lines 205 to 206)

“Although the formation and structure of B27 and B28 are thought to be rational, the reaction mechanism from, for example, B13-2 to B13-3 is not yet clear.”

Comment 2.

Page 3 line 71-77: Please reword this paragraph and move the comments linked to figure 2 to section 2 (Results & Discussion). Additionally, move the figure to a more appropriate section of the text and include with more precision and detail the objective of the article.

Response 2.

We move Figure 2 to “Results & Discussion” section.

In addition, the text was changed.

(lines 74 to 75)

“In this study, we assigned the structures of the product ions generated from the [M+H]+ ion of naturally occurring 51-hydroxy CTX3C [1] using LC-APCI-QTOFMS (Figure 2) based on CRF [15, 16].”

=> “In this study, we assigned the structures of the product ions generated from the [M+H]+ of naturally occurring 51-hydroxyCTX3C [1] based on CRF [17, 18].”

Reviewer 3 Report

Comments and Suggestions for Authors

In this manuscript, authors gave a detail analysis generating product ions of 51-hydroxyCTX3C, which was reasonable. And the drawings were nice, although some figures were too small to see them clearly when printed. The reported mechanisms were interesting and important, which made this work worth publishing in this journal.

However, minor revisions were required.

1. Had the induce charge-remote fragmentation (CRF) been found in other types of polycyclic ethers before?

2. Had the inductive effect of a hydroxy substituent in the structures of polycyclic ethers been reported before?

3. Figures 3 and 4 were too small to see them clearly. Please also check Figures S2–S4 and S8–S10. They were blurry to see data clearly.

Other revisions:

1. P1L15: ‘on C51induced’ → ‘on C51 induced’

2. P1L22: ‘51hydroxyCTX3C’ → ’51-hydroxyCTX3C’

3. Figure 2 caption: superscript: ‘[M+H]+’ → ‘[M+H]+’. Please check the Figures S2–S4 and S8–S10 captions, too.

4. P3L90: ‘be-tween’ → ‘between’

5. P3L97: ‘Figure S1 and S2’ → ‘Figures S1 and S2’. Please check the similar typo errors throughout the whole manuscript, such as ‘Figure S1 and S3’, ‘Figure 3 (a), 7 (a), and S1’.

Comments on the Quality of English Language

There were a few grammar or typo errors. Some of them were given in the comments to the authors.

Author Response

Thank you for your peer review to our manuscript.

We have revised the manuscript ID: toxins-2813151 on the basis of the Referee’s comments.

We look forward to a publication of our manuscript in the toxins.

Reviewer 3

Minor revisions

Comment 1.

Had the induce charge-remote fragmentation (CRF) been found in other types of polycyclic ethers before?

Response 1.

Yes, it is known. For example, in the MS/MS spectrum of brevetoxin, it was also observed in the continuous dehydration commonly observed in polycyclic ethers. Although there are no reported mechanisms for this, it is expected to occur based on CRF as well as CTX.

We added References 14 and 15 to line 69.

  1. “Abraham A, Wang Y, Said R.E K, Plakas M. S, Characterization of brevetoxin metabolism in Karenia brevis bloom-exposed clams (Mercenaria sp.) by LC-MS/MS. Toxicon, 2012, 60, 1030–1040.”
  2. “Liu X, Ma Y, Wu J, Yin Q, Wang P, Zhu J, Chan L L, Wu B. Characterization of New Gambierones Produced by Gambierdiscus balechii 1123M1M10. marine drugs, 2023, 21, 3.”

Comment 2.

Had the inductive effect of a hydroxy substituent in the structures of polycyclic ethers been reported before?

Response 2.

To the best of our knowledge, there have been no reports regarding this so far. Our present report is the first on the inductive effect of hydroxy substituents on the structure of polycyclic ethers.

Comment 3.

Figures 3 and 4 were too small to see them clearly. Please also check Figures S2–S4 and S8–S10. They were blurry to see data clearly.

Response 3.

We changed layout and enlarged Figures 3 and 4 to its maximum size. Furthermore, we changed the figure of Figures S2-S4 and S8-S10.

Other revisions

Comment 1.

P1L15: ‘on C51induced’ → ‘on C51 induced’

Response 1.

(line 15) ‘on C51induced’ => “on C51 induced”

Comment 2.

P1L22: ’51hydroxyCTX3C’ → ‘51-hydroxyCTX3C’

Response 2.

(line 22) ’51hydroxyCTX3C’ => “51-hydroxyCTX3C”

Comment 3.

Figure 2 caption: superscript: ‘[M+H]+’ → ‘[M+H]’. Please check the Figures S2–S4 and S8–S10 captions, too.+

Response 3.

(Figure 2 caption: superscript and Figures S2-S4 and S8-S10, and line 226) “[M+H]+” => “[M+H]+

Comment 4.

P3L90: ‘be-tween’ → ‘between’

Response 4.

(line 92) ‘be-tween’ => ‘between’

Comment 5.

P3L97: ‘Figure S1 and S2’ → ‘Figures S1 and S2’. Please check the similar typo errors throughout the whole manuscript, such as ‘Figure S1 and S3’, ‘Figure 3 (a), 7 (a), and S1’.

Response 5.

(line 99) ‘Figure S1 and S2’ => “Figures S1 and S2”